# The Effect of Titanium Oxide Additions on the Phase Chemistry and Properties of Chromite-Magnesia Refractories

**Johan PR De Villiers, Delphin Mulange and Andrie Mariana Garbers-Craig** *

Department of Materials Science & Metallurgical Engineering, University of Pretoria, Pretoria 0002, South Africa; johan.devilliers@up.ac.za (J.P.D.V.); delmulange@gmail.com (D.M.)
* Correspondence: andrie.garbers-craig@up.ac.za; Tel.: +27-12-420-3189

**Abstract:** The microstructure of a direct-bonded chromite-magnesia refractory brick, typically used in copper and platinum converters, was modified by adding different amounts of nano-size $TiO_2$ to the raw material mixture. Bricks with 0, 1, 3, 5, and 7 mass% $TiO_2$ were produced and compared in terms of spinel formation; the role of the tetravalent cation $Ti^{4+}$ in the bonding phase; as well as changes in density, porosity, thermal expansion, and internal stress. This was done through a comprehensive XRD and SEM-EDS study. It was found that Ti is accommodated in the secondary spinel that has formed, where Mg in excess of unity in the tetrahedral site combines with an equal amount of Ti in the octahedral sites to maintain charge balance. The 1 mass% $TiO_2$ brick had the lowest bulk density (but not significantly different from the original chromite-magnesia brick), the smallest difference in unit cell volumes between the primary and secondary spinels, and the lowest stress arising from the smallest difference in linear thermal expansion coefficients of the phases present. The calculated porosities correspond well with experimentally determined apparent porosity values, whereas the linear thermal expansion coefficients calculated at 1392K are similar to the values measured from 293 to 1273 K.

**Keywords:** chromite-magnesia brick; refractory; spinel; $TiO_2$

## 1. Introduction

Magnesia-chromite and chromite-magnesia bricks are mainly used in the non-ferrous industry (in smelters and converters used in platinum group metal (PGM), copper, nickel, and lead extraction), but also in vacuum degassers in the steel industry and CLU (Creusot-Loire-Uddeholm) converters in the ferroalloy industry [1–4]. These bricks have the benefit of enhanced thermal shock resistance as compared to magnesia-based materials, as well as high corrosion resistance with respect to slightly acidic to basic slags, specifically to fayalitic slags that are rich in both silica and iron oxide [1–3]. Magnesia-chromite and chromite-magnesia refractories are categorized in silicate, direct, and rebonded grades based on the bonding system between the grains [5]. The silicate phase within the refractory matrix facilitates liquid phase sintering at relatively low temperatures but deteriorates the thermomechanical properties of the refractory material in high-temperature applications [6]. High-duty direct-bonded bricks with low amounts of silicate impurities are therefore used when excellent refractoriness is required at high temperatures. Previous work has shown that direct bonding and the use of fused magnesia-chromite grains are essential in improving the properties of the bricks. Direct bonds form through secondary spinel formation between the magnesia and magnesia-chromite grains [7,8]. Three types of chromite grains and crystals can typically be distinguished in as-received direct-bonded magnesia-chromite bricks: blended primary chromite (called Type I), secondary chromite

crystals present as a rim around blended chromite and/or at magnesia grain boundaries (Type 2), and secondary chromite crystals that have exsolved out of magnesia (Type 3) [8]. Chromite ore consists mainly of the complex solid solution phase $(Mg,Fe^{2+})(Cr,Al,Fe^{3+})_2O_4$, which has a cubic spinel structure with structural formula unit $M_{3-\delta}O_4$, where $M = Mg^{2+}$, $Fe^{2+}$, $Cr^{3+}$, $Al^{3+}$, and $Fe^{3+}$, and $\delta$ indicates the deviation from stoichiometry [9]. When $\delta$ is positive, cation vacancies associated with internal oxidation of $Fe^{2+}$ to $Fe^{3+}$ are present, whereas when $\delta$ is negative, interstitials are present.

Higher operating temperatures in PGM smelters over the last few decades, which are due to increased amounts of chromium in the slag, have resulted in increased matte superheat, which implies that highly fluid matte is produced [10]. To improve the matte penetration resistance of magnesia-chromite and chromite-magnesia bricks, the bricks should be further densified without compromising their thermal shock resistance. Therefore, promoting the proportion of direct bonding between refractory aggregate grains by redesigning the microstructure of chromite-magnesia refractories with particular focus on the formation of secondary spinel phases could make these materials more penetration-resistant. Studies have reported the improvement of sintering through homogeneous dispersion of nano-sized particles in the microstructure of the refractory matrix and subsequent spinel formation [11–16]. Tetravalent cation oxides such as $TiO_2$ and $SnO_2$ are known to greatly improve the spinel sintering process as they decrease the synthesis temperature, and also increase the thermodynamic stability of the spinel solid solution phase [17]. A comparative study on the densification of magnesia-rich ceramics using $Ti^{4+}$ and $Sn^{4+}$ has shown that spinels containing $Ti^{4+}$ are denser than those containing $Sn^{4+}$. Furthermore, the addition of appropriate amounts of $TiO_2$, generally up to 2%, can significantly increase spinel densification [18]. This is due to the formation of a single-phase spinel solid solution with the other spinels, whereas spinels containing $Sn^{4+}$ do not form a single-phase solid solution [14].

In this study, the impact of micron-size $TiO_2$ additions on phase evolution in the microstructure of a chromite-magnesia brick, which is typically used in copper and platinum converters, was investigated. This was done through a detailed XRD and SEM-EDS study on the phases that formed when 0, 1, 3, 5, and 7 mass% $TiO_2$ were, respectively, added to the chromite-magnesia raw material mixture. The relation between the formation of secondary spinels with varying amounts of $TiO_2$ and the density, porosity, percentage linear expansion, and stress build-up in chromite-magnesia-based test bricks was investigated.

## 2. Materials and Methods

### 2.1. Preparation of the Brick Samples

The starting materials for preparation of the chromite-magnesia-based bricks were composed of fused grains (prepared by blending magnesite and chromite and fusing at 2000 °C), chromite ore, and two types of dead burnt magnesia (DBM X1 and DBM X2). Molasses and dextrin were used as binders. For each composition, apart from the reference batch, high-purity industrial-grade $TiO_2$ provided by Chemgrit SA (Pty) Limited was substituted at weight percentages of 1, 3, 5, and 7. Five batches labeled CMDi were prepared, where "i" refers to the mass% of substituted $TiO_2$. For each batch, $TiO_2$ was added in such a way that it substitutes the same weight percentages of the sum of the ball mill fractions of dead burnt magnesia and chromite ore (Table 1).

The $-4 + 2$ and $-2 + 0$ mm size fraction starting raw materials along with half of the total amount of $TiO_2$ were thoroughly mixed at room temperature in a Hobart mixer for 20 min to ensure homogeneity. Powdered dextrin and molasses solution were then added to the mixture and mixed for another 20 min, and then the fine burnt magnesia, chromite, and the remaining $TiO_2$ were added to the mixture. The total mixture was then mixed for another 30 min. Each batch was then uniaxially pressed at 1.8 ton/cm$^2$ into bricks (230 mm × 114 mm × 76 mm), which were then dried at 110 °C for 24 h before being kept at 200 °C in the preheating zone of a tunnel kiln for 48 h. The brick samples were subsequently fired in the tunnel kiln between 1690 and 1700 °C at a push rate of 16 cars per day, and a soaking time of 3 h at

the peak temperature. All refractory samples were prepared at Vereeniging Refractories (Pty) Limited (South Africa).

**Table 1.** Raw material compositions of the bricks.

| Material | Screen Size (mm) | CMD0 | CMD1 | CMD3 | CMD5 | CMD7 |
|---|---|---|---|---|---|---|
| | | wt% | wt% | wt% | wt% | wt% |
| Fused grain | −4 + 2 | 30 | 30 | 30 | 30 | 30 |
| Fused grain | −2 + 0 | 18 | 18 | 18 | 18 | 18 |
| Chromite ore | −2 + 0 | 14 | 14 | 14 | 14 | 14 |
| Chromite ore | Ball mill | 5 | 4.5 | 3.5 | 2.5 | 1.5 |
| DBM X1 | −2 + 0 | 10 | 10 | 10 | 10 | 10 |
| DBM X2 | Ball mill | 20 | 19.5 | 18.5 | 17.5 | 16.5 |
| $TiO_2$ | <150 μm | 0 | 1 | 3 | 5 | 7 |
| Molasses | | 3 | 3 | 3 | 3 | 3 |
| Dextrin | | 0.8 | 0.8 | 0.8 | 0.8 | 0.8 |

*2.2. Mineralogical Analysis of the Bricks*

Portions of each refractory brick were cut and pulverized to provide fine-grained material for chemical and phase characterization. For XRD analysis, the samples were ground to a smaller particle size (<15 microns) using a McCrone micronizing mill. Polished sections were prepared for optical and scanning electron microscopy. The latter samples were coated with gold to make them conductive.

2.2.1. Bulk Chemical Analysis

The chemical analyses were performed by UIS Analytical Services using Inductively Coupled Plasma Optical Emission Spectrometry (ICP-OES) analysis of all major elements. These are expressed as oxide percentages, including total Fe as $Fe_2O_3$.

2.2.2. XRD Analysis

The XRD measurements were done on a PANalytical X-ray diffractometer equipped with a fast Xcelerator detector, whereas phase quantification was performed using the TOPAS™ Rietveld program [19]. Crystal structures of the phases to be used in Rietveld analysis were obtained from the Crystallography Open Database (COD). In addition, the site occupancies of the spinels and periclase were modified to correspond with the SEM analytical data. The resultant calculated densities of the different phases were subsequently used to calculate their volume percentages and bulk densities. The perfect cleavage of the periclase resulted in the preferred orientation, which was corrected (partially) by a March–Dollase correction in the (001) orientation [20]. No preferred orientation corrections of the other phases were attempted because of the wide variation in the quantification results when this was tried. $R_{wp}$ values varied between 4.2% and 6%, which was considered acceptable.

High-temperature X-ray diffraction measurements were taken (in air) using an Anton Paar heating chamber equipped with a platinum heating strip. Temperature calibration was done using a pure MgO external standard from 800 to 1500 °C with the thermal expansion data provided by Touloukian et al. [21]. Thermal expansion was measured by the determination of the unit cell expansion of all phases.

2.2.3. SEM Analysis

Energy dispersive analysis of all phases was performed using a JEOL IT300 scanning electron microscope equipped with a XMAX50 Si-drifted EDS detector. Standardless analyses were done on spectra collected at 60 s counting times. All elements are expressed as oxide percentages, assuming standard valencies for the elements, including trivalent Fe. These were then used to calculate the structural formulas of each of the phases in the different samples. The very fine-grained spinels (in sample CMD0) that exsolved from the periclase (secondary chromite Type 3 [8]) were analyzed

separately because they are usually as big as or smaller than the interaction volume of the electron beam, resulting in a composite analysis of the inclusion and the host periclase.

Because of the relatively high errors associated with standardless analysis using Li-drifted EDS detectors, some microprobe standards were analyzed using the Si-drifted detector used in all the analysis reported in this study. The results are given in Table S1 of the Supplementary Information. This included the analysis of the oxygen concentrations. In general, the deviations from the accepted values vary below 1.7% absolute and 4% relative. These are considered to be acceptable for the purposes of this investigation.

### 2.3. Bulk Density and Apparent Porosity Measurements

Bulk densities and apparent porosities of five specimens from each composition were measured at room temperature using water immersion porosimetry (WIP) based on ISO 5017 and DIN 51918. Average values were reported. The WIP method measures the porosity and the bulk density by saturating a sample with a liquid of known density, and calculating the pore volume from the weight difference between the fully saturated and dry states. The total volume of the sample is determined by immersing it in the fluid using Archimedes' Principle. Cylindrical specimens, 50 mm in diameter and 50 mm in height, were first dried at 110 °C before being tested.

### 2.4. Thermal Expansion Coefficient

The linear coefficients of thermal expansion of the produced bricks were determined using the BS 1902:5.3:1900 standard method. Measurements were performed on dried cylindrical samples of 25.4 ± 0.15 mm diameter and 12 mm in height, under argon, using an Orton dilatometer in the temperature range from 20 to 1000 °C.

## 3. Results

### 3.1. Bulk Chemical Analysis

The starting materials were first characterized in terms of chemical composition as listed in Table 2. The MgO content of the bricks varied between 44.0 and 45.6 mass%, and the $Cr_2O_3$ content between 22.3 and 26.4 mass%.

**Table 2.** Average chemical compositions of brick CMDi and the titania used (mass%) [1].

| | $SiO_2$ | $Al_2O_3$ | $Fe_2O_3$ | $TiO_2$ | CaO | MgO | $Cr_2O_3$ | Total |
|---|---|---|---|---|---|---|---|---|
| CMD0 | 2.34 | 9.65 | 12.60 | 0.28 | 0.77 | 44.65 | 26.40 | 96.69 |
| CMD1 | 2.31 | 8.72 | 12.11 | 1.21 | 0.89 | 45.55 | 25.79 | 96.58 |
| CMD3 | 1.81 | 8.03 | 12.35 | 3.88 | 0.65 | 45.35 | 23.45 | 95.52 |
| CMD5 | 2.05 | 8.40 | 11.67 | 4.88 | 0.79 | 45.25 | 23.89 | 96.91 |
| CMD7 | 2.93 | 8.48 | 12.03 | 6.87 | 0.99 | 44.00 | 22.33 | 97.63 |
| Titania | 0.13 | 1.22 | 0.06 | 97.46 | 0.09 | - | - | 98.96 |

[1] Average of two ICP-OES analyses.

### 3.2. Microscopy

The microstructures of the samples are similar as they consist of three principal spinel types together with periclase (MgO) and minor amounts of forsterite ($Mg_2SiO_4$) and monticellite ($CaMgSiO_4$). This is shown in Figure 1a,b. Spinel type A is prismatic in shape and retains this shape as the $TiO_2$ content increases (primary spinel, Type 1 [8]). Spinel B is irregular in shape and appears to segregate to the periclase grain boundaries and grow irregularly in size as the $TiO_2$ content increases (secondary spinel, Type 2 [8]). Spinel C (secondary spinel, Type 3 [8]), which exsolved from the periclase (grouped together with spinel B), also decreases in size as the $TiO_2$ content increases. Minor amounts of forsterite and monticellite occur on the grain boundaries between spinel A and periclase.

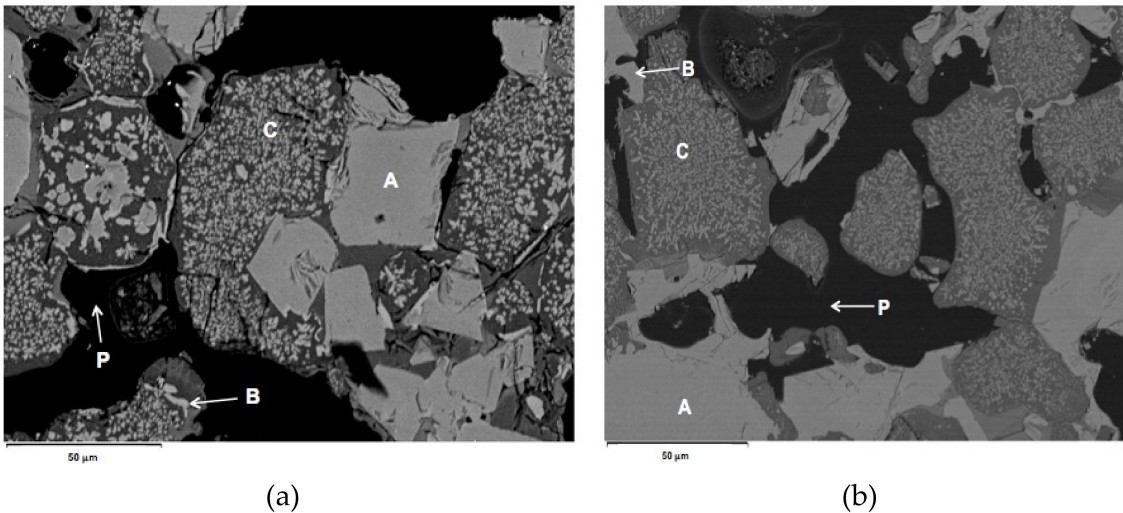

(a)                                                                (b)

**Figure 1.** Electron backscatter images of samples CMD0 (**a**) and CMD3 (**b**). The prismatic relict spinel (A) is contrasted with the irregularly shaped spinel (B) associated with the periclase (P). Spinel C is present as very fine-grained inclusions in periclase.

### 3.3. Phase Chemistry

The phase compositions are shown as mass% in Table 3 and as phase formulas in Table 4. The latter are shown as the average structural formulas normalized to three cations and four oxygen atoms, together with their standard deviations. This normalization method was used to calculate the $Fe^{2+}$ and $Fe^{3+}$ contents, similar to the method used by Quintiliani et al. [22] and Ferracutti et al. [23]. An example of the method is given as Table S2 in the Supplementary Information. The periclase compositions were normalized to one cation. From the formula calculations, the Fe in the spinels appears to be predominantly trivalent.

From the analyzed spinel compositions, it is apparent that two major spinel types are present in all of the CMD samples (apart from the fine-grained exsolved spinel present in sample CMD0). In Tables 3 and 4, the spinel compositions labeled as spinel A represent the compositions of the coarse relict original chrome spinel. The spinels labeled as spinel B are the spinels that reacted with the magnesia and $TiO_2$ to form new spinels with increased Ti contents. The fine-grained inclusions in the periclase show higher iron contents than those of the coarser spinel B inclusions and are labeled as spinel C. The compositions of spinels B and C (in sample CMD0) were grouped together to reconcile them with the XRD analyses (which show only two distinct spinels) and with the ICP analyses.

In all samples, the Mg occupancy in the relict chrome spinel A is unity (based on three cations) within its standard deviation (see Table 4). This means that the tetrahedral sites are fully occupied with Mg, and that the octahedral sites must be exclusively occupied by the trivalent cations due to the octahedral site preference of Cr and Al.

The Mg contents in the secondary B spinels increase concurrently with the increase in Ti content. This is apparent as the Mg occupancies that are in excess of 1.0 are equal to their Ti contents (within the errors of analysis). For example, spinel B in sample CMD3 has an Mg content of 1.27 atoms per formula unit and a Ti content of 0.22 atoms per formula unit. This means that the $Mg^{2+}$ is in excess of 1.0 resides in the octahedral site and is balanced by an equal amount of octahedral $Ti^{4+}$ to have an effective valency of $(Mg + Ti)^{3+}$, resulting in a spinel that is charge-balanced. This holds for all newly formed spinel B grains that were analyzed. The difference in Ti content of spinels A and B in sample CMD3 is apparent in their EDS spectra shown in Figure S1 in the Supplementary Information.

**Table 3.** Average compositions of the spinels and periclase (mass%) in samples containing 0, 1%, 3%, 5%, and 7% $TiO_2$ [1,2].

**CMD0**

| Analysis | Spinel A Average | SD | Spinel B Average | SD | Spinel C Average | SD | Periclase Average | SD |
|---|---|---|---|---|---|---|---|---|
| $TiO_2$ | 0.45 | 0.14 | 0.50 | 0.30 | 0.00 | 0.00 | | |
| $Al_2O_3$ | 21.74 | 5.68 | 22.19 | 0.72 | 6.33 | 1.06 | | |
| $Cr_2O_3$ | 48.21 | 6.56 | 47.11 | 1.10 | 22.08 | 4.39 | 1.98 | 0.69 |
| $Fe_xO$ [1] | 6.54 | 0.96 | 6.18 | 0.39 | 42.59 | 2.99 | 5.25 | 1.81 |
| MgO | 23.05 | 1.44 | 24.01 | 0.88 | 28.99 | 5.21 | 92.77 | 2.50 |
| SUM | 100.00 | | 100.00 | | 100.00 | | 100.00 | |
| N [2] | 4 | | 5 | | 13 | | 5 | |

**CMD1**

| Analysis | Spinel A Average | SD | Spinel B Average | SD | Periclase Average | SD |
|---|---|---|---|---|---|---|
| $TiO_2$ | 2.97 | 1.93 | 4.69 | 0.27 | | |
| $Al_2O_3$ | 13.67 | 0.79 | 12.07 | 0.32 | | |
| $Cr_2O_3$ | 44.47 | 8.30 | 30.85 | 3.88 | 2.66 | 0.11 |
| $Fe_xO$ [1] | 15.55 | 2.57 | 26.28 | 3.85 | 5.41 | 1.28 |
| MgO | 23.34 | 3.63 | 26.11 | 0.25 | 91.93 | 1.37 |
| SUM | 100.00 | | 100.00 | | 100.00 | |
| N [2] | 4 | | 8 | | 5 | |

**CMD3**

| Analysis | Spinel A Average | SD | Spinel B Average | SD | Periclase Average | SD |
|---|---|---|---|---|---|---|
| $TiO_2$ | 1.06 | 0.79 | 10.61 | 1.93 | | |
| $Al_2O_3$ | 15.05 | 1.78 | 10.34 | 1.14 | | |
| $Cr_2O_3$ | 52.48 | 3.86 | 22.77 | 3.58 | 2.15 | 0.73 |
| $Fe_xO$ [1] | 8.32 | 4.43 | 27.22 | 2.28 | 4.74 | 1.11 |
| MgO | 23.09 | 0.59 | 29.07 | 1.27 | 93.10 | 1.71 |
| SUM | 100.00 | | 100.01 | | 99.99 | |
| N [2] | 5 | | 9 | | 7 | |

**CMD5**

| Analysis | Spinel A Average | SD | Spinel B Average | SD | Periclase Average | SD |
|---|---|---|---|---|---|---|
| $TiO_2$ | 0.49 | 0.49 | 13.34 | 1.69 | | |
| $Al_2O_3$ | 18.18 | 3.29 | 11.11 | 1.09 | | |
| $Cr_2O_3$ | 50.88 | 2.07 | 22.10 | 4.01 | 2.14 | 0.19 |
| $Fe_xO$ [1] | 7.89 | 3.11 | 23.29 | 2.55 | 5.80 | 0.89 |
| MgO | 22.57 | 1.40 | 29.98 | 0.95 | 92.06 | 1.04 |
| SUM | 100.00 | | 99.82 | | 100.00 | |
| N [2] | 4 | | 9 | | 3 | |

**CMD7**

| Analysis | Spinel A Average | SD | Spinel B Average | SD | Periclase Average | SD |
|---|---|---|---|---|---|---|
| $TiO_2$ | 0.18 | 0.36 | 16.75 | 2.98 | | |
| $Al_2O_3$ | 18.83 | 3.57 | 9.98 | 1.61 | | |
| $Cr_2O_3$ | 52.92 | 1.76 | 22.72 | 7.75 | 2.16 | 0.39 |
| $Fe_xO$ [1] | 5.87 | 3.39 | 19.26 | 6.67 | 2.61 | 0.51 |
| MgO | 22.21 | 1.24 | 31.26 | 1.74 | 95.23 | 0.36 |
| SUM | 100.00 | | 99.97 | | 100.00 | |
| N [2] | 4 | | 5 | | 2 | |

[1] In spinels the Fe oxide is shown as $Fe_2O_3$ and in periclase as FeO. [2] N is the number of analyses contributing to the averages.

**Table 4.** Average formulas based on 3 cations per formula unit of spinels and periclase in samples containing 0, 1%, 3%, 5%, and 7% $TiO_2$.

| | CMD0 | | | | | | | | | |
|---|---|---|---|---|---|---|---|---|---|---|
| | Spinel A | | Spinel B | | Spinel C | | Periclase | | | |
| Analysis | Average | SD | Average | SD | Average | SD | Average | SD | | |
| Ti | 0.010 | 0.003 | 0.011 | 0.007 | 0.002 | 0.005 | | | | |
| Al | 0.740 | 0.168 | 0.753 | 0.020 | 0.280 | 0.158 | | | | |
| Cr | 1.110 | 0.179 | 1.072 | 0.033 | 0.599 | 0.238 | 0.022 | 0.008 | | |
| Σ Fe | 0.143 | 0.023 | 0.134 | 0.009 | 0.850 | 0.303 | 0.066 | 0.022 | | |
| Mg | 0.997 | 0.037 | 1.030 | 0.030 | 1.270 | 0.188 | 0.912 | 0.017 | | |
| SUM | 3.000 | | 3.000 | | 3.000 | | 1.000 | | | 3 |
| N | 4 | | 5 | | 13 | | 5 | | | |

| | CMD1 | | | | | | | CMD3 | | | | | | |
|---|---|---|---|---|---|---|---|---|---|---|---|---|---|---|
| | Spinel A | | Spinel B | | Periclase | | | | Spinel A | | Spinel B | | Periclase | |
| Analysis | Average | SD | Average | SD | Average | SD | | Analysis | Average | SD | Average | SD | Average | SD |
| Ti | 0.062 | 0.049 | 0.105 | 0.007 | | | | Ti | 0.024 | 0.018 | 0.220 | 0.066 | | |
| Al | 0.524 | 0.069 | 0.423 | 0.010 | | | | Al | 0.528 | 0.060 | 0.349 | 0.036 | | |
| Cr | 1.027 | 0.170 | 0.726 | 0.078 | 0.030 | 0.001 | | Cr | 1.236 | 0.085 | 0.565 | 0.163 | 0.030 | 0.008 |
| Σ Fe | 0.301 | 0.135 | 0.592 | 0.076 | 0.068 | 0.016 | | Σ Fe | 0.187 | 0.101 | 0.601 | 0.048 | 0.059 | 0.016 |
| Mg | 1.086 | 0.053 | 1.154 | 0.015 | 0.902 | 0.017 | | Mg | 1.026 | 0.030 | 1.265 | 0.065 | 0.911 | 0.023 |
| SUM | 3.000 | | 3.000 | | 1.000 | | | SUM | 3.000 | | 3.000 | | 1.000 | |
| N | 4 | | 8 | | 5 | | | N | 5 | | 9 | | 7 | |

| | CMD5 | | | | | | | CMD7 | | | | | | |
|---|---|---|---|---|---|---|---|---|---|---|---|---|---|---|
| | Spinel A | | Spinel B | | Periclase | | | | Spinel A | | Spinel B | | Periclase | |
| Analysis | Average | SD | Average | SD | Average | SD | | Analysis | Average | SD | Average | SD | Average | SD |
| Ti | 0.014 | 0.010 | 0.293 | 0.037 | | | | Ti | 0.004 | 0.008 | 0.365 | 0.063 | | |
| Al | 0.653 | 0.104 | 0.389 | 0.038 | | | | Al | 0.654 | 0.115 | 0.341 | 0.052 | | |
| Cr | 1.174 | 0.063 | 0.503 | 0.098 | 0.025 | 0.002 | | Cr | 1.235 | 0.053 | 0.521 | 0.178 | 0.021 | 0.005 |
| Σ Fe | 0.169 | 0.081 | 0.514 | 0.058 | 0.074 | 0.013 | | Σ Fe | 0.131 | 0.079 | 0.421 | 0.148 | 0.040 | 0.009 |
| Mg | 0.991 | 0.054 | 1.301 | 0.039 | 0.901 | 0.015 | | Mg | 0.977 | 0.047 | 1.351 | 0.060 | 0.940 | 0.004 |
| SUM | 3.000 | | 3.000 | | 1.000 | | | SUM | 3.000 | | 3.000 | | 1.000 | |
| N | 4 | | 9 | | 3 | | | N | 4 | | 5 | | 2 | |

The average Ti occupancy in the secondary spinels (B) gradually increases from 0.011 to 0.365 (based on three cations) as the $TiO_2$ addition increases from zero to 7 percent. The Ti occupancy of the relict spinels varies randomly between 0.004 and 0.057. This clearly indicates that the relict spinels react with the $TiO_2$ to a limited extent. This is shown in Figure 2. Likewise, the periclase compositions do not change significantly as $TiO_2$ is added to the refractory mix.

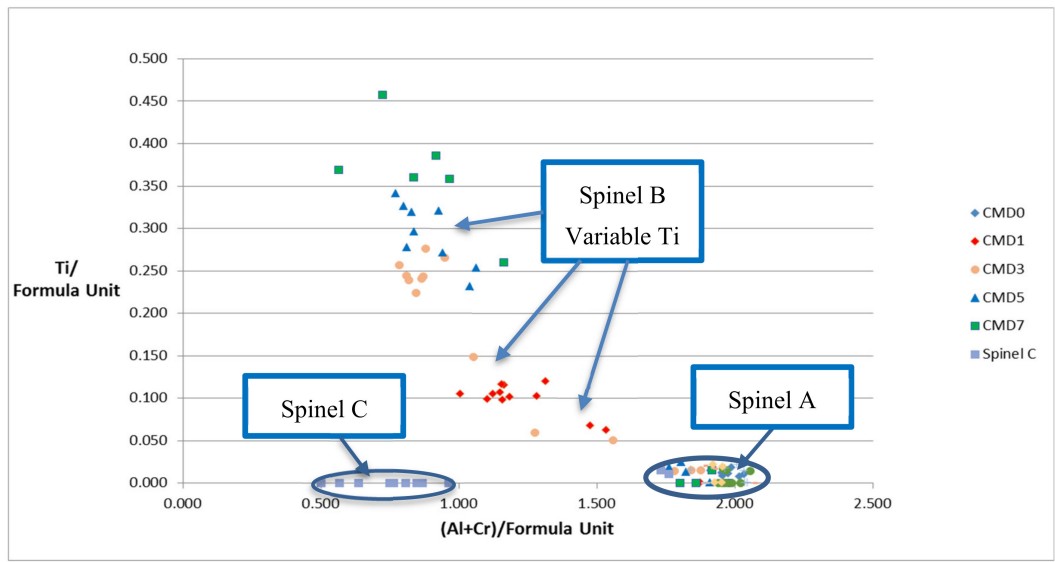

**Figure 2.** Compositional data for the analyzed spinels showing the increase in Ti content (per formula unit) of the secondary spinel B. The Ti contents of the relict spinel A are near zero for all $TiO_2$ additions in the samples. Spinel C inclusions in periclase (in CMD0) are also devoid of Ti.

*3.4. Mass and Volume Percentages of the Phases*

The variation in phase contents and unit cell volumes are given in Table 5. Using XRD, only two spinel types show clearly resolved peaks, and they were quantified as spinels A and B (see Figure 3). For each of the samples, the average elemental occupancies given in Table 4 were used in the Rietveld refinement. The preferred orientation of the periclase affected the accuracy of the phase quantities but the values given in Table 5 are the averages of four different XRD determinations (two different samples mounted and measured twice). This enabled the calculation of accurate densities as well as the volume percentages of the phases in the different samples.

**Table 5.** Quantities (mass%) and unit cell volumes ($Å^3$) of spinels A and B, periclase, and minor constituents of monticellite and forsterite. Standard deviations of the phase quantities are given in parentheses.

|  | Spinel A | | Spinel B | | MgO | | Monticellite | | Forsterite | |
|---|---|---|---|---|---|---|---|---|---|---|
|  | Mass% | Vol ($Å^3$) | Mass% | Vol ($Å^3$) | Mass% | Vol ($Å^3$) | Mass% | Vol ($Å^3$) | Mass% | Vol ($Å^3$) |
| CMD0 | 40(3) | 565.1 | 27(5) | 581.3 | 24(8) | 74.8 | 4(1) | 336.0 | 6(1) | 294.0 |
| CMD1 | 21(1) | 569.2 | 42(1) | 576.0 | 28(2) | 74.7 | 4(1) | 336.3 | 4(1) | 293.1 |
| CMD3 | 24(1) | 568.6 | 46(1) | 581.2 | 25(2) | 74.7 | 2(1) | 337.1 | 3(1) | 293.2 |
| CMD5 | 24(1) | 568.8 | 48(3) | 582.6 | 21(4) | 74.8 | 3(1) | 337.1 | 4(1) | 293.2 |
| CMD7 | 28(2) | 568.4 | 40(1) | 588.2 | 22(1) | 74.8 | 3(1) | 337.4 | 7(1) | 293.1 |

The Rietveld results of CMD0, CMD1, and CMD3 are shown in Figure 3 together with the peak allocations of spinel A, spinel B, and periclase. The correspondence between the calculated and experimental diffraction data is given in the difference curve (bottom curve). The small difference in the unit cells of spinels A and B in CMD1 is apparent in their almost coincident peaks at 75 degrees 2-theta.

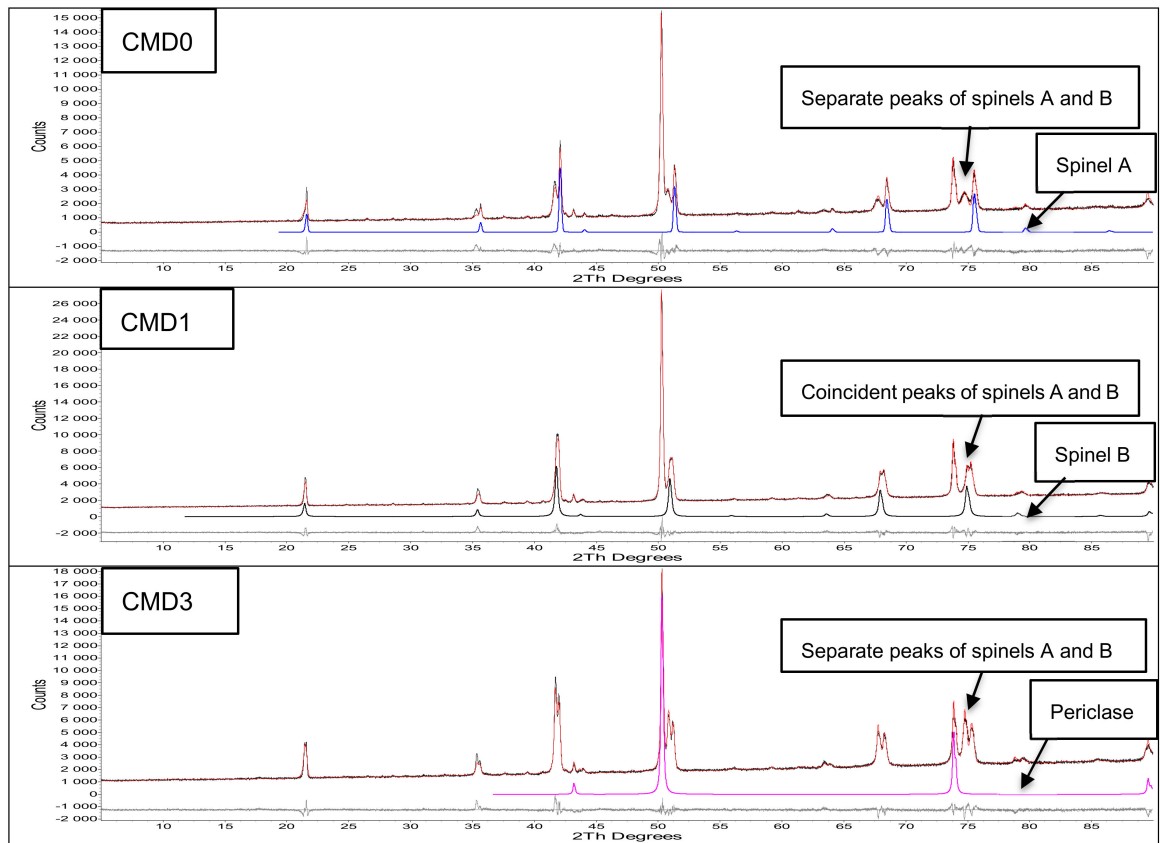

**Figure 3.** Results of the Rietveld refinement of samples CMD0, CMD1, and CMD3. The positions of the peaks (in degrees 2θ) of spinel A, spinel B, and periclase are shown for cobalt CoKα radiation. The bottom difference curve shows the correspondence between the calculated and experimental intensity data.

A mass balance calculation was done to compare the compositions calculated from the XRD phase quantification (Table 5) and the average phase compositions in Table 3 with that of the ICP analysis given in Table 1. The comparison is given in Table S3 in the Supplementary Information. Obviously, this is dependent on the accuracy of the XRD analysis (2% absolute) and whether the phase compositions are representative, based on limited sampling. The $TiO_2$ and $Fe_2O_3$ compositions calculated from XRD and EDS analysis result in overdetermination in the samples, whereas the MgO and $Cr_2O_3$ values are comparable.

The unit cell volumes of spinels A and B are the most similar in CMD1 (Table 5). The cell volumes of spinel A are similar in samples CMD1 to CMD7, whereas those of spinel B gradually increase with higher substitutions of $TiO_2$. The differences in unit cell dimensions between spinels A and B in CMD0 to CMD7 are 0.0785 Å, 0.0328 Å, 0.0605 Å, 0.0663 Å, and 0.0949 Å, respectively.

Because the volume percentages of the phases do not differ markedly from the mass percentages, and to illustrate the volume changes in the samples, the volume fractions were also calculated. These relate to the volumes per 100 g sample and were calculated by dividing the mass percentages of each phase by their respective densities. The latter were calculated directly from the average formulas and the refined lattice parameters, and will differ from sample to sample for each phase. This is shown in Table 6, and the respective mass percentages, volume fractions of the individual phases, and total volume changes are shown in Figure 4b.

**Table 6.** Calculated densities (g/cm$^3$), volume fractions, and volume percentages of the different phases in samples CMD0 to CMD7. The sum value gives the total volume change per 100 g sample.

| CMD0 | | | | | | |
|---|---|---|---|---|---|---|
| | **Spinel A** | **Spinel B** | **Periclase** | **Monticellite** | **Forsterite** | **Sum** |
| Density | 4.08 | 4.00 | 3.72 | 3.17 | 3.18 | |
| Vol fract | 9.798 | 6.667 | 6.377 | 1.217 | 1.825 | 25.885 |
| Vol% | 37.85 | 25.76 | 24.63 | 4.70 | 7.05 | 100.000 |

| CMD1 | | | | | | |
|---|---|---|---|---|---|---|
| | **Spinel A** | **Spinel B** | **Periclase** | **Monticellite** | **Forsterite** | **Sum** |
| Density | 4.16 | 4.13 | 3.84 | 3.16 | 3.19 | |
| Vol fract | 5.14 | 10.28 | 7.37 | 1.21 | 1.27 | 25.27 |
| Vol% | 20.35 | 40.70 | 29.15 | 4.78 | 5.02 | 100.00 |

| CMD3 | | | 3 | | | |
|---|---|---|---|---|---|---|
| | **Spinel A** | **Spinel B** | **Periclase** | **Monticellite** | **Forsterite** | **Sum** |
| Density | 4.20 | 4.05 | 3.81 | 3.16 | 3.19 | |
| Vol fract | 5.79 | 11.29 | 6.49 | 0.66 | 1.00 | 25.23 |
| Vol% | 22.93 | 44.74 | 25.74 | 2.62 | 3.97 | 100.00 |

| CMD5 | | | 5 | | | |
|---|---|---|---|---|---|---|
| | **Spinel A** | **Spinel B** | **Periclase** | **Monticellite** | **Forsterite** | **Sum** |
| Density | 4.14 | 3.99 | 3.85 | 3.16 | 3.19 | |
| Vol fract | 5.89 | 12.05 | 5.56 | 0.80 | 1.13 | 25.44 |
| Vol% | 23.16 | 47.39 | 21.86 | 3.14 | 4.45 | 100.00 |

| CMD7 | | | 7 | | | |
|---|---|---|---|---|---|---|
| | **Spinel A** | **Spinel B** | **Periclase** | **Monticellite** | **Forsterite** | **Sum** |
| Density | 4.15 | 3.93 | 3.75 | 3.15 | 3.19 | |
| Vol fract | 6.81 | 10.20 | 5.88 | 0.84 | 2.19 | 25.91 |
| Vol% | 26.28 | 39.37 | 22.68 | 3.23 | 8.44 | 100.00 |

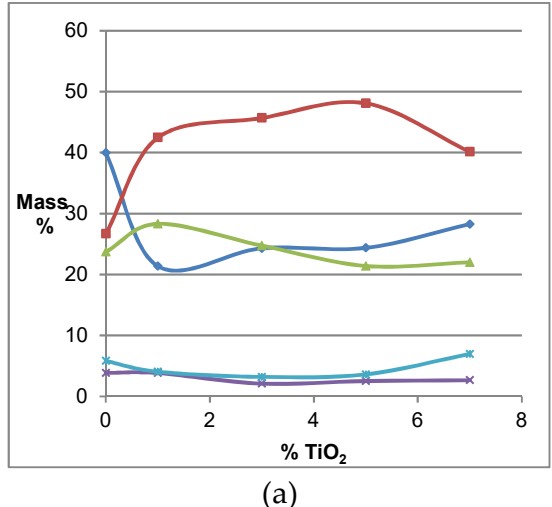
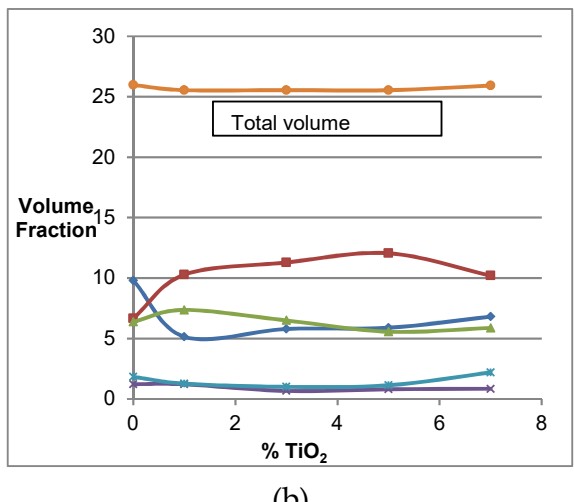

(a)      (b)

**Figure 4.** (**a**) Mass percentages of the phases in sample CMDi (0 to 7% TiO$_2$): spinel B (■, red), periclase (▲, green), spinel A (♦, blue), forsterite (✱, light blue), and monticellite (✖, purple). (**b**) Individual volume fractions and total volume changes. Total volume change (●, brown).

### 3.5. Bulk Density and Porosity

The theoretical bulk densities $\rho_t$ of the samples were calculated using their mass percentages and respective densities:

$$\rho_t = \sum_i X_i \rho_i \tag{1}$$

where $X_i$ is the mass fraction of the phase and $\rho_i$ is its density.

The bulk density and apparent porosity values of five specimens from each composition were measured using the standard water immersion method, and average values were taken. The theoretical bulk densities were compared with the measured bulk densities, and from these the porosities were calculated using the following formula,

$$\varnothing = 1 - \frac{\rho_b}{\rho_t} \tag{2}$$

where $\rho_b$ is the measured bulk density and $\rho_t$ is the theoretical density.

The calculated porosities were then compared to the measured apparent porosities. These comparisons are shown in Table 7. The calculated porosities agree well with the measured ones, with CMD0 and CMD1 slightly underdetermined.

**Table 7.** Comparison of theoretical bulk densities with observed values together with derived and measured apparent porosities.

| Sample | Bulk Density (g/cm$^3$) | | Porosity (%) | |
|---|---|---|---|---|
| | Theoretical | Measured | Calculated | Measured |
| CMD0 | 3.89 | 3.29 | 15.3 | 16.4 |
| CMD1 | 3.87 | 3.19 | 17.6 | 18.4 |
| CMD3 | 3.91 | 3.07 | 21.4 | 20.7 |
| CMD5 | 3.91 | 2.91 | 25.5 | 25.1 |
| CMD7 | 3.88 | 2.81 | 27.6 | 27.2 |

The theoretical bulk density of CMD1 is the minimum compared to the other compositions, but not significantly different from the other samples.

### 3.6. Thermal Expansion

Theoretical composite linear thermal expansion coefficients were calculated using the formula given by Turner [24], which was used by Dealmeida [25] for ceramic materials and for complex heterogeneous materials, as summarized by Karch [26]:

$$\alpha_{comp} = \frac{\sum_i (\alpha_i K_i F_i / \rho_i)}{\sum_i (K_i F_i / \rho_i)} \tag{3}$$

where $\alpha_i$ is the coefficient of linear thermal expansion of phase $i$, $K_i$ is the bulk modulus of phase $i$, $F_i$ is the mass fraction of phase $i$, and $\rho_i$ is the density of phase $i$.

The standard formula for the calculation of $\alpha_i$ was used:

$$\alpha_i = \frac{1}{L_0} \frac{\partial L}{\partial T} \tag{4}$$

where $L_0$ is the lattice parameter of phase $i$ at 298 K.

Because both monticellite and forsterite are minor components, only the linear expansions of their intermediate crystallographic axes were measured and used in the calculation of the bulk expansion coefficients.

The bulk moduli of the spinels were calculated using the formula given by Zhongying et al. [27] that relates the bulk moduli of the spinel end members to the bulk modulus of the solid solution:

$$K_{ss} = \sum_i x_i K_i \tag{5}$$

where $K_{ss}$ is the bulk modulus of the solid solution and $K_i$ is the bulk modulus of the spinel end members.

The spinel end members of the spinels in samples CMD0 to CMD7 were calculated using the end member generator (EMG) (Ferracutti et al. [23]) and the oxide analyses as determined by EDS analysis. These are given in Table 8, together with their calculated bulk moduli. The bulk moduli of the end members of $MgAl_2O_4$, $MgFe_2O_4$, and $MgCr_2O_4$ were taken from Zhongying et al. [27] and that of $Mg_2TiO_4$ from Mingda et al. [28]. For periclase and forsterite, the bulk moduli were taken from Shanker et al. [29], and for monticellite, from Sharp et al. [30]. The densities of the phases were determined in this study.

**Table 8.** End member compositions of the A and B spinel solid solutions (normalized to 8) and their calculated bulk moduli (in GPa).

| Spinel | $MgAl_2O_4$ | $MgFe_2O_4$ | $MgCr_2O_4$ | $Mg_2TiO_4$ | $K_{ss}$ |
|---|---|---|---|---|---|
| CMD0-A | 2.971 | 0.571 | 4.419 | 0.039 | 187.2 |
| CMD1-A | 1.977 | 1.436 | 4.314 | 0.274 | 185.0 |
| CMD3-A | 2.141 | 0.756 | 5.008 | 0.096 | 185.7 |
| CMD5-A | 2.867 | 0.589 | 4.447 | 0.097 | 187.0 |
| CMD7-A | 2.589 | 0.515 | 4.88 | 0.016 | 186.6 |
| CMD0-B | 3.058 | 0.544 | 4.355 | 0.044 | 187.4 |
| CMD1-B | 1.838 | 2.555 | 3.151 | 0.456 | 184.1 |
| CMD3-B | 1.662 | 2.794 | 2.456 | 1.088 | 183.1 |
| CMD5-B | 1.802 | 2.412 | 2.405 | 1.381 | 183.2 |
| CMD7-B | 1.656 | 2.041 | 2.529 | 1.774 | 182.7 |

The individual linear thermal expansion coefficients were determined from high-temperature powder XRD measurements of the lattice parameters of the individual phases as determined by Rietveld analysis. The temperatures of the experiments were calibrated using pure MgO periclase as an external standard and calibration every 100 degrees, using the thermal expansion data of Touloukian et al. [21].

The thermal expansion calibration (lattice constant vs. temperature for MgO) is given in Figure S2a, and the actual temperature calibration (sample temperature vs. instrument reference temperature) is given in Figure S2b in the Supplementary Information file.

The lattice parameter data of the refractory phases were fitted to a second-order polynomial and the linear thermal expansion coefficients determined by differentiating the polynomial at two temperatures, 1392 K and 1840 K.

The individual and composite linear thermal expansion coefficients are tabulated in Table 9 for the two temperatures mentioned. Because of the predominance of the spinel and periclase phases, these phases, and especially the spinel B phase, affect the bulk or composite thermal expansion coefficients the most. The expansion coefficients of the minor phases monticellite and forsterite are not very well determined because of the errors in lattice constant refinements. The linear expansion coefficients for all the phases are shown in Table 9 and Figure 5, and for the composite samples ($\alpha_{comp}$) at two temperatures, in Table 9 and Figure 6. The actual thermal expansion data as well as their second-order polynomial fits are given as Figure S3 in the Supplementary Information. The reason for calculating $\alpha_{comp}$ at 1392 K and 1840 K was to calculate it at higher temperatures than 1273 K (1000 °C), which is a low temperature for refractory usage.

**Table 9.** Individual and composite linear thermal expansion coefficients derived for samples CMD0 to CMD7. All values in $K^{-1}$.

| Phase | CMD0 | | CMD1 | | CMD3 | | CMD5 | | CMD7 | |
|---|---|---|---|---|---|---|---|---|---|---|
| | $\alpha i$ (1392) | $\alpha i$ (1840) | $\alpha i$ (1392) | $\alpha i$ (1840) | $\alpha i$ (1392) | $\alpha i$ (1840) | $\alpha i$ (1392) | $\alpha i$ (1840) | $\alpha i$ (1392) | $\alpha i$ (1840) |
| Spinel A | $9.15 \times 10^{-6}$ | $1.09 \times 10^{-5}$ | $8.86 \times 10^{-6}$ | $1.16 \times 10^{-5}$ | $1.01 \times 10^{-5}$ | $1.47 \times 10^{-5}$ | $8.15 \times 10^{-6}$ | $1.29 \times 10^{-5}$ | $7.86 \times 10^{-6}$ | $1.01 \times 10^{-5}$ |
| Spinel B | $1.43 \times 10^{-7}$ | $-2.2 \times 10^{-5}$ | $8.81 \times 10^{-6}$ | $3.44 \times 10^{-6}$ | $8.46 \times 10^{-6}$ | $2.65 \times 10^{-7}$ | $8.19 \times 10^{-6}$ | $-1.5 \times 10^{-7}$ | $8.61 \times 10^{-6}$ | $-6.1 \times 10^{-6}$ |
| Periclase | $1.33 \times 10^{-5}$ | $7.22 \times 10^{-6}$ | $1.33 \times 10^{-5}$ | $9.67 \times 10^{-6}$ | $1.4 \times 10^{-5}$ | $7.74 \times 10^{-6}$ | $1.3 \times 10^{-5}$ | $1.13 \times 10^{-5}$ | $1.29 \times 10^{-5}$ | $1.19 \times 10^{-5}$ |
| Monticellite | $1.54 \times 10^{-5}$ | $-2.9 \times 10^{-5}$ | $4.89 \times 10^{-6}$ | $-1.01 \times 10^{-4}$ | $1.6 \times 10^{-6}$ | $1.72 \times 10^{-5}$ | $1.18 \times 10^{-5}$ | $1.55 \times 10^{-5}$ | $5.22 \times 10^{-6}$ | $-2 \times 10^{-6}$ |
| Forsterite | $3.37 \times 10^{-6}$ | $8.28 \times 10^{-6}$ | $5.46 \times 10^{-6}$ | $8.07 \times 10^{-7}$ | $1.58 \times 10^{-5}$ | $1.53 \times 10^{-5}$ | $5.53 \times 10^{-6}$ | $3.32 \times 10^{-6}$ | $3.79 \times 10^{-6}$ | $2.19 \times 10^{-5}$ |
| $\alpha$comp | $7.47 \times 10^{-6}$ | $-4.8 \times 10^{-7}$ | $9.8 \times 10^{-6}$ | $3.54 \times 10^{-6}$ | $1.03 \times 10^{-5}$ | $6.31 \times 10^{-6}$ | $9.14 \times 10^{-6}$ | $5.83 \times 10^{-6}$ | $8.93 \times 10^{-6}$ | $4.23 \times 10^{-6}$ |

In Figure 5, the linear expansion coefficients of spinel B are markedly different from the others, because it becomes negative in both CMD0 and CMD7 and levels off in the other samples. The reason for the decreases in thermal expansion coefficients of spinel B is most likely that the inclusions of spinel B in periclase are affected by the one order of magnitude larger thermal expansion coefficient of periclase. This results in compression of the spinel B inclusions.

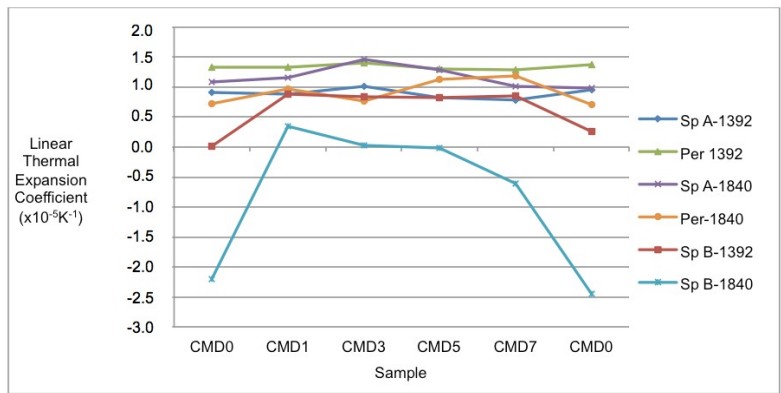

**Figure 5.** Individual linear thermal expansion coefficients (in $K^{-1}$) for spinel phases A and B as well as periclase at 1392 and 1840 K in samples CMD0 to CMD7, with a re-analysis of CMD0. The coefficients for spinel B at 1840 K (especially in sample CMD0) are significantly different from the coefficients for spinel B at 1392 K and the other phases at the two temperatures.

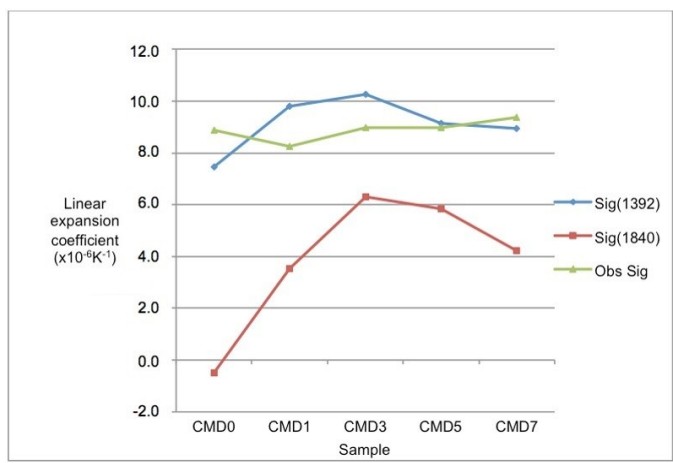

**Figure 6.** Calculated linear expansion coefficients for samples CMD0 to CMD7 at 1392 and 1840 K. Values are in $K^{-1}$. These are compared with the measured expansion coefficients taken from 293 to 1273 K.

The linear thermal expansion coefficient of spinel B also has the highest value in sample CMD1 and is closest to the band defined by the other phases. This implies that the stress that develops in the refractory when the temperature increases will be the lowest for sample CMD1, due to the smallest differences in linear expansion coefficients.

The composite linear expansion coefficients for the samples are shown in Figure 6. The 1392 K calculated values are similar to the values measured from 293 to 1273 K using the BS 1902-5.3:1900 method. Detailed differences are probably because porosity was not taken into account in the calculations.

To determine the stresses resulting from the difference in thermal expansion, the formula derived by Turner [22] was used:

$$S_i = (\beta_r - \beta_i)\Delta T K_i \tag{6}$$

where $\beta_r$ and $\beta_i$ are the volume expansion coefficients ($\beta = 3\alpha$ for cubic compounds) of the total sample and of the individual phases, respectively; $\Delta T$ is the temperature interval; and $K_i$ is the bulk modulus of phase $i$. The results are shown in Figure 7. The stress calculation was done for only the two spinels as well as for periclase in the temperature interval 1392–1840 K. This was because of the predominance of these phases in the samples and the fact that the volume expansion coefficients of monticellite and forsterite are poorly defined.

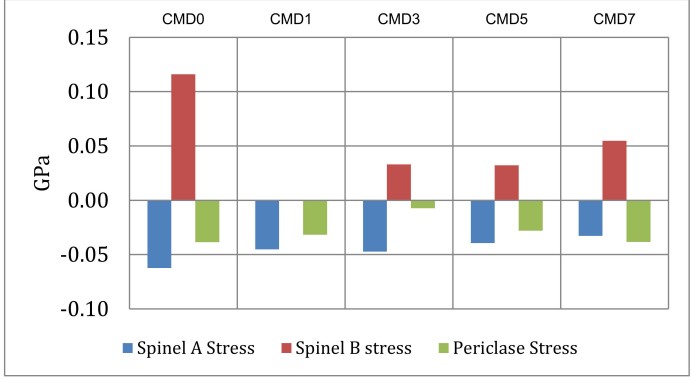

**Figure 7.** Calculated stress difference between the volume expansion coefficients of the individual phases and the composite samples. The difference between spinel B and the bulk in sample CMD1 is negligibly small.

## 4. Discussion

The energy dispersive analyses clearly show that the Ti is exclusively incorporated in the secondary spinel B that is primarily associated with the periclase. From Table 4, the Ti content of spinel B (based on 3 cations) increases from 0.011 in CMD0 to 0.365 in CMD7. This is contrasted with the Ti content of relict spinel A, which varies randomly between 0.004 and 0.041. It can also be observed that addition of Ti enables the incorporation of Mg into the octahedral sites of spinel B. There is a clear correspondence between the Ti content and the Mg content in excess of unity (Table 10). This shows that the excess Mg is situated in the octahedral sites to ensure charge balance in the spinel. The octahedral sites are now occupied by cations $Al^{3+}$, $Fe^{3+}$, and $(Mg + Ti)^{3+}$, where the $Mg^{2+}$ and $Ti^{4+}$ must be equal to maintain the trivalent charge and for the spinel to be charge-balanced. The excess of Mg in spinel B is provided by the periclase, and it is reflected by a decrease in the periclase content when spinel B increases with increased Ti substitution (Table 4).

The XRD analysis shows an initial sharp increase in the content of secondary spinel B, up to a maximum of 48%, followed by a gradual decrease as the $TiO_2$ content further increases. There is a concomitant decrease in spinel A as the $TiO_2$ content increases. Periclase initially increases and then gradually decreases with increasing $TiO_2$ content.

Comparing the unit cell volumes of spinels A and B with different $TiO_2$ substitutions in Table 11, it is apparent that the difference in unit cell volumes between spinels A and B is the minimum in sample CMD1. The reason for this difference is not known but could be due to a smaller stress concentration in this sample.

**Table 10.** Comparison between the Ti occupancy and the Mg occupancy in excess of 1.

|  | CMDO | CMD1 | CMD3 | CMD5 | CMD7 |
|---|---|---|---|---|---|
| Ti occupancy | 0.011 | 0.105 | 0.220 | 0.293 | 0.365 |
| Excess Mg | 0.030 | 0.154 | 0.265 | 0.301 | 0.351 |

**Table 11.** Difference in unit cell volumes between spinels A and B with increased $TiO_2$ substitution.

|  | Spinel A Vol($\text{Å}^3$) | Spinel B Vol($\text{Å}^3$) | % Difference |
|---|---|---|---|
| CMD0 | 565.1 | 581.3 | 2.87 |
| CMD1 | 569.2 | 576.0 | 1.19 |
| CMD3 | 568.6 | 581.2 | 2.22 |
| CMD5 | 568.8 | 582.6 | 2.43 |
| CMD7 | 568.4 | 588.2 | 3.48 |

Using the chemical formulas and the unit cell volumes, the theoretical densities were calculated. These are shown for every phase in every sample in Table 6. For example, the density of spinel B varies from 3.93 to 4.13 in the samples. The volume change (volume in $cm^3$ per 100 g sample) can be calculated from the mass percentages and the densities of the individual phases. This shows that the volume changes of the different phases follow their mass percentages. More importantly, the volume change after firing stays essentially the same.

The calculated porosities of the samples were compared to the measured apparent porosities and, in general, there is good correspondence, with the calculated values of CMD0 and CMD1 slightly underestimated compared to the observed values.

The calculation of the linear thermal expansion coefficients, especially at high temperatures, shows a trend towards negative expansion coefficients or a flattening of the expansion of spinel B, which is accentuated by the increase in its content in all samples except sample CMD0. This decrease in expansion coefficients is most likely due to compression of the spinel inclusions by the host periclase. The difference in the expansion coefficients is the smallest in sample CMD1, and the stress due to this difference is also the smallest in this sample. This is graphically shown in Figure 7 for the temperature interval 1392–1840 K.

## 5. Conclusions

The impact of adding micro-size $TiO_2$ to a chromite-magnesia brick formulation was examined in terms of phase formation and how it impacts on the physical and thermal properties of the brick. We draw the following conclusions.

The examined bricks, which contained either 0, 1, 3, 5, or 7 mass% $TiO_2$, consisted of three spinel types (primary spinel A; secondary spinel B, which segregated to the periclase boundaries; and secondary spinel C that exsolved from the periclase) together with periclase and minor amounts of forsterite and monticellite. The added $TiO_2$ is accommodated in the secondary spinel B. Mg is increasingly partitioned in the octahedral sites of the secondary spinel, with an accompanying decrease in the amounts of relict spinel A, and to a lesser extent in the periclase, with increasing $TiO_2$ content. The Mg in excess of unity in the tetrahedral site combines with an equal amount of Ti in the octahedral sites to maintain charge balance. The difference in unit cell volumes between spinels A and B is the minimum when 1 mass% $TiO_2$ is added.

The bulk densities and porosities of the different bricks were calculated from XRD data and compared with measured data. The calculated porosities of the different samples correspond well with the experimental apparent porosity values. Both theoretical and measured density and porosity data indicated that $TiO_2$ addition to the chromite-magnesia brick did not result in a significant densification of the brick.

The linear thermal expansion coefficients of spinel B show a surprising decrease to negative values or a leveling off with increasing temperature. The stress arising from differences in thermal expansion

coefficients of the phases present compared to that of the composite sample is the minimum in sample CMD1 (1 mass% $TiO_2$). Comparing the composite calculated linear thermal expansion coefficients at 1392 K with those measured up to 1273 K shows good correspondence. The values at 1840 K are dramatically lower because of the decreased or negative linear expansion coefficients of spinel B.

Although the examined chromite-magnesia bricks could not meaningfully be densified by replacing chromite and magnesia in the ball mill fraction of the raw material mixture with 1–7 mass% micron-size $TiO_2$, the addition of 1 mass% $TiO_2$ was beneficial in terms of lowering the stress that is associated with the $TiO_2$-free chromite-magnesia brick.

The derivation of useful properties of magnesia-chromite refractories from microchemical and crystallographic evaluation is demonstrated in this study. These methods could be useful in the evaluation of existing refractories and could also be a useful tool in the design of novel formulations. Aspects such as volume changes upon heating, the effect of minor element additions, and comparison of thermal expansion coefficients can be evaluated before these materials are produced commercially.

**Supplementary Materials:** The following are available online at http://www.mdpi.com/2571-6131/3/1/13/s1. Table S1. Standardless SEM analysis of microprobe standards. Table S2. Formula calculation of Spinel A in sample CMD0. Table S3. Comparison between bulk compositions calculated from XRD and EDS analysis and ICP analysis. Figure S1. EDS spectra of Spinel A (top) and spinel B (bottom) from sample CMD3. Figure S2. (a) Variation of the periclase lattice parameter with temperature from the data of Toloukian et al. [21] and (b) the corrected sample temperature as calibrated from high temperature XRD data of MgO. Figure S3. Thermal expansion data for CMD samples. All diagrams show lattice constant as a function of temperature in degrees Kelvin.

**Author Contributions:** Conceptualization, J.P.d.V. and A.M.G.-C.; preparation of brick samples, measurement of bulk densities, apparent porosities, and linear thermal expansion, D.M.; interpretation of XRD data, J.P.d.V.; writing of paper, J.P.d.V. and A.M.G.-C.; project administration and funding acquisition, A.M.G.-C. All authors have read and agreed to the published version of the manuscript.

**Funding:** This research was funded through the THRIP (Technology and Human Resources for Industry Program) of South Africa.

**Acknowledgments:** The assistance and comments of Volker Kahlenberg are greatly appreciated. The accurate XRD analysis, also at high temperatures, by Wiebke Grote is also gratefully acknowledged.

**Conflicts of Interest:** The authors declare no conflicts of interest.

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
