# Peer review of "The Effect of Titanium Oxide Additions on the Phase Chemistry and Properties of Chromite-Magnesia Refractories"

_ceramics, doi:10.3390/ceramics3010013_

Round 1
Reviewer 1 Report
The aim of this work is to present the impact of TiO2 addition on phase evolution in the microstructure of a chromite-magnesia brick. The issue is important from the application point of view. For this purpose, the authors used two research techniques: XRD and SEM-EDS. The relationship between forming of secondary spinels (with varying amounts of TiO2) on structure and some physical properties of bricks was studied.
Samples preparation has been well described. According to the reviewer, the most important for the article is Table 3, where compositions of the spinels are presented. I am seriously concerned that the values ​​presented in Table 3 are in fact subject to significantly greater measurement uncertainty than the given values. It is necessary to confirm that the authors have taken into account the limitations of the EDS technique. Without this, the rest of the article has no value.
Please, present EDS spectra and quantitative analysis results (it can be in Supplementary Materials). In the analysis, please take into account the fact that for major elements (20-5%) relative accuracy for quantitative standardless analysis is 4% (for minor elements 10-20%; for trace 50%). Please describe also oxygen (light element) content determination.
Explain the standardization method used in order to calculate Fe content (page 4, line 136). Figure 2 is unclear. The axes should be described. Please cut the unnecessary digits (for example 2.5 instead of 2.500). I suggest that you do not label samples with different Ti content. Instead, it's better to label Spinel A, Spinel B and Periclase differently. Similarly, all other drawings need to be improved.
If the authors show that the participation of Mg and Ti was determined correctly, the impact of TiO2 addition on phase evolution will be convincing (as well as the interpretation that addition of Ti enables the incorporation of Mg into the octahedral sites of spinel).
Author Response
- I am seriously concerned that the values presented in Table 3 are in fact subject to significantly greater measurement uncertainty than the given values. It is necessary to confirm that the authors have taken into account the limitations of the EDS technique. Without this, the rest of the article has no value.
The standardless EDS analyses were varified using microprobe standards. This is described in lines 120 – 124 and the data given in Table A, Supplementary information.
- Please, present EDS spectra and quantitative analysis results (it can be in Supplementary Materials). In the analysis, please take into account the fact that for major elements (20-5%) relative accuracy for quantitative standardless analysis is 4% (for minor elements 10-20%; for trace 50%). Please describe also oxygen (light element) content determination.
EDS spectra of spinels A and B from CMD3 are shown in Figure A, Supplementary Information.
- Explain the standardization method used in order to calculate Fe content (page 4, line 136).
This is described in lines 145 – 147. An example calculation is given in Table B, Supplementary Information.
- Figure 2 is unclear. The axes should be described.
Done.
- Please cut the unnecessary digits (for example 2.5 instead of 2.500).
Normally compositions are given to two decimals and unit cell contents to three decimals, since they are normalised to only three atoms per formula unit. Detail might be lost if it is only given to one or two decimals.
- I suggest that you do not label samples with different Ti content. Instead, it's better to label Spinel A, Spinel B and Periclase differently.
The authors believe that since the types of spinels are similar in the different samples, the spinel names (Spinel A, Spinel B and Spinel C) should be kept consistent and the sample names should be changed according to the amount of TiO2 that was added. The different spinels have been more clearly described in section 3.3 (lines 150-156 were added).
- Similarly, all other drawings need to be improved.
Attention was given to improve all the Figures. The axes were also more clearly labelled.
Reviewer 2 Report
The authors mentioned that the study was done through detailed XRD and SEM-EDS analyses, but the reviewer cannot find any XRD pattern or EDS spectra in the paper. How can the readers confirm the change of composition in the bricks?
More details of the experiment measuring the thermal expansion coefficient (TEC) should be provided. Here the reviewer has some questions for the authors: Why did the authors compare TEC measured at 293-1273K with a TEC calculated at different temperatute (1392 K)? In addition, Is the method mentioned in reference [22] used for plastic, not for refractory materials?
English and especially format editing is strongly recommended before the re-submission. Most of the graphs (and some tables) need to be revised because of: lack of axis title, the format of the number (no trailing zero after the decimal point), unnecessary graph title... The font size of equations is also needed to be unified.
The Conclusions should be rewritten more coherently and logically. Now, it looks like a list of results without emphasis on what is the important finding of this research? what can we do to further develop this study or apply it?
Author Response
- The authors mentioned that the study was done through detailed XRD and SEM-EDS analyses, but the reviewer cannot find any XRD pattern or EDS spectra in the paper. How can the readers confirm the change of composition in the bricks?
XRD (Figure 3) and SEM-EDS (Figure A, Supplementary Information) spectra were added.
- More details of the experiment measuring the thermal expansion coefficient (TEC) should be provided.
Section 2.4 was added in which the experimental determination of the TEC is described.
- Here the reviewer has some questions for the authors: Why did the authors compare TEC measured at 293-1273K with a TEC calculated at different temperatute (1392 K)?
The reason was motivated in lines 293-294. The TEC was calculated at a higher temperature as it is more applicable to the temperature range in which these refractory bricks are used.
- Is the method mentioned in reference [22] used for plastic, not for refractory materials?
No, it is also used for ceramic materials and complex heterogeneous materials. – This comment was added to section 3.6.
- Most of the graphs (and some tables) need to be revised because of: lack of axis title, the format of the number (no trailing zero after the decimal point), unnecessary graph title... The font size of equations is also needed to be unified.
Done.
- The Conclusions should be rewritten more coherently and logically. Now, it looks like a list of results without emphasis on what is the important finding of this research?
Conclusions were rewritten.
Reviewer 3 Report
l.126 please check if a word is missing in the sentence „… in that…“
sections 3.2 and 3.3: why did the authors use 2 naming systems for the spinels (spinel 1 = type A, spinel 2 = type B). Either both names are used every time or only one denomination is used throughout the text.
Section 3.2./Figure 1: where is the 3rd spinel type in Figure 1?
- 138 ff: the relation between the example and Table 3 is not clear. How does Table 3 give the oxidic compositions?
Figure 1: In the SEM images are some marking (probably from edx analyses) which are not mentioned in the text.
Table 3: how are the compositions given in the table? Do the data show mass fractions or atom fractions? For the evaluation of spinel A as “relict chrome spinel” and spinel B as secondary spinel a comparison the chemical composition of the raw materials would be interesting.
Page 7: page number is not 7 but 1
Figure 2: What are the units of the shown data?
Table 4: Vol(ų) seems to be the unit cell volume (according to the text) - this should be noted in the caption
l. 186: why do the cell volumes of spinel A and B differ so much in CMD0 although the chemical compositions are so similar in comparison the to the differences between the two spinels in the other samples? This should be discussed.
l. 189: why does CMD1 show the smallest difference in unit cell dimensions?
Table 5: What is the difference between volume fraction and Vol%? Usually, the volume fraction multiplied by 100% gives volume percent. Apparently, the authors do not refer to this. They might refer to the volume change (Figure 3b) – But this needs to be described and named more clearly
Figure 3: Please add titles to the x axes
l.205: Please check formula 1 – rho_i should be in the numerator
- 212: Please discuss the closed porosity. The closed porosity cannot be measured by immersion techniques
- 299: please give a reference to a relavant table or figure showing this observation
- 302 ff: the authors should discuss this effect more detailed – how can this effect be possibly explained?
l.310: please check the example values from table 5 – the density values for spinel B vary from 3.93 to 4.13
Author Response
- l.126 please check if a word is missing in the sentence „… in that…“
Corrected.
- Sections 3.2 and 3.3: why did the authors use 2 naming systems for the spinels (spinel 1 = type A, spinel 2 = type B). Either both names are used every time or only one denomination is used throughout the text.
The authors used the A and B naming system, while the 1 and 2 naming system is from Goto and Lee, reference [8].
- Section 3.2./Figure 1: where is the 3rdspinel type in Figure 1?
Spinel C was also labelled in Figures 1a and 1b.
- 138 ff: the relation between the example and Table 3 is not clear. How does Table 3 give the oxidic compositions?
Corrected.
- Figure 1: In the SEM images are some marking (probably from edx analyses) which are not mentioned in the text.
Removed.
- Table 3: how are the compositions given in the table? Do the data show mass fractions or atom fractions? For the evaluation of spinel A as “relict chrome spinel” and spinel B as secondary spinel a comparison the chemical composition of the raw materials would be interesting.
‘Mass %’ was added to the heading of Table 3. The average chemical compositions of the different spinels are given in Table 3, while their stoichiometries are given in Table 4 (this is an added Table).
- Page 7: page number is not 7 but 1
Corrected.
- Figure 2: What are the units of the shown data?
The axes were properly labeled.
- Table 4: Vol(ų) seems to be the unit cell volume (according to the text) - this should be noted in the caption
Corrected.
- l. 186: why do the cell volumes of spinel A and B differ so much in CMD0 although the chemical compositions are so similar in comparison the to the differences between the two spinels in the other samples? This should be discussed.
The composition of spinel C is now included, which changes the overall composition of CMD0.
- l. 189: why does CMD1 show the smallest difference in unit cell dimensions?
The reason is not known, but this could be due to the smallest difference in stress.
- Table 5: What is the difference between volume fraction and Vol%? Usually, the volume fraction multiplied by 100% gives volume percent. Apparently, the authors do not refer to this. They might refer to the volume change (Figure 3b) – But this needs to be described and named more clearly
The Vol% was calculated from the individual volume fractions of each phase divided by the sum of volume fractions, multiplied by 100. The calculation of the volume fractions are described in lines 222-226.
- Figure 3: Please add titles to the x axes
Corrected.
- l.205: Please check formula 1 – rho_i should be in the numerator
Corrected, thank you.
Round 2
Reviewer 2 Report
1. XRD (Figure 3) and SEM-EDS (Figure A, Supplementary Information) spectra were added.
additional comment: in line 229, please correct the element in "CoKα". Please increase the size of the legend. If possible, please take the raw data and draw the pattern using some graph software like Excel. Also, the intensity (or count) axis should be presented in arbitrary unit (arb.u.)
The size of characters in Figure A (Supplementary) should be larger.
2. Section 2.4 was added in which the experimental determination of the TEC is described
additional comment: the description is clear.
3. The reason was motivated in lines 293-294. The TEC was calculated at a higher temperature as it is more applicable to the temperature range in which these refractory bricks are used.
additional comment: the answer is reasonable.
4. No, it is also used for ceramic materials and complex heterogeneous materials. – This comment was added to section 3.6.
additional comment: the answer is clear.
5. Though the graphs and tables are well edited, but there are some details can be done to improve the quality of the figures. For example, in Figure 7, Gpa (horizontally arranged) should be GPA (vertically arranged). In Figure B and C (Supplementary), the axis titles and the unit should be added. The function of the trend lines can be removed (they are too long and cross the trend lines in some figures), just show the R-square.
6. Conclusions were rewritten --> Yes, now the reviewer can understand the conclusions of the authors better.
Author Response
- In line 229, please correct the element in "CoKα".
‘Cobalt’ was added in front of the CoKa.
- Please increase the size of the legend. If possible, please take the raw data and draw the pattern using some graph software like Excel. Also, the intensity (or count) axis should be presented in arbitrary unit (arb.u.)
The legends on Figure 3 were omitted because they do not add any additional information to what was supplied in the Tables. The purpose of the XRD traces is to show the peaks of the major components and the close coincidence of the spinel peaks in CMD1.
- The size of characters in Figure A (Supplementary) should be larger. Corrected.
- In Figure 7, Gpa (horizontally arranged) should be GPA (vertically arranged).
Corrected.
- In Figure B and C (Supplementary), the axis titles and the unit should be added. The function of the trend lines can be removed (they are too long and cross the trend lines in some figures), just show the R-square.
Corrected.